# Conventional Transarterial Chemo embolization Using Streptozocin in Patients with Unresectable Neuroendocrine Liver Metastases

**DOI:** 10.3390/cancers15164021

**Published:** 2023-08-08

**Authors:** Audrey Fohlen, Remi Beaudouin, Arnaud Alvès, Karine Bouhier-Leporrier, Christophe Pasik, Jean-Pierre Pelage

**Affiliations:** 1Interventional Radiology, Caen University Medical Center, 14033 Caen, France; audrey.fohlen.2410@gmail.com (A.F.); remi.al.beaudouin@gmail.com (R.B.); 2Centre National de la Recherche Scientifique, Imaging & Therapeutic Strategies for Cancer & Brain Tissue UMR 6030 GIP CYCERON “ISTCT-CERVOxy”, Normandie Caen University, 14000 Caen, France; 3Department of Digestive Surgery, Caen University Medical Center, 14033 Caen, France; alvesar@hotmail.com; 4Interdisciplinary Research Unit for Cancer Prevention and Treatment “ANTICIPE”, Inserm Unity UMR 1086, Normancy Caen University, Calvados General Tumor Registry, Centre François Baclesse, 14000 Caen, France; 5Department of Hepato-Gastroenterology and Digestive Oncology, Caen University Medical Center, 14033 Caen, France; bouhierleporrier-k@chu-caen.fr; 6Riemser Pharma, 92120 Montrouge, France

**Keywords:** liver metastases, neuroendocrine tumor, transarterial chemoembolization, streptozocin

## Abstract

**Simple Summary:**

Neuroendocrine tumors are frequently associated with liver metastases at the time of diagnosis. Trans-catheter arterial embolization with conventional chemoembolization is one of the reference palliative treatment in patients with uncontrolled carcinoid syndrome or progressive disease. The aim of our study was to evaluate the tolerability and clinical, biological and radiological tumor response and survival rates in patients with unresectable neuroendocrine liver metastases treated by trans-catheter arterial embolization with conventional chemoembolization, using streptozocin, Lipiodol and embolization microspheres. At the end of 127 procedures, carcinoid syndrome was improved in 69% of patients after treatment; objective response and non-progressive disease were 32% and 70%, respectively. The OS at 1 year, 2 years, 3 years and 5 years was 91% (IC = 84–99%), 84% (CI = 72–95%), 69% (CI = 53–84%) and 63% (C = 46–81%), respectively. This study suggests that this procedure using streptozocin is an effective and well-tolerated palliative option for patients with unresectable neuroendocrine liver metastases, which can be repeated and induces durable response and disease control

**Abstract:**

Background: The purpose of this study was to evaluate the clinical, biological and radiological responses to, and tolerability of, conventional transarterial chemoembolization (cTACE) using streptozocin for unresectable neuroendocrine liver metastases. Patients and Methods: A total of 52 patients with predominant liver disease were treated with cTACE using an emulsion of streptozocin, Lipiodol and embolization particles. A sequential approach was favored in patients with high liver tumor burden. Clinical, biological and radiological responses were evaluated using carcinoid symptoms, biomarkers and mRecist criteria, respectively. Results: A total of 127 procedures were performed with a sequential approach in 65% of patients. All patients received streptozocin and Lipiodol. Carcinoid syndrome was improved in 69% of patients after treatment (*p* = 0.01). Post-embolization syndrome was reported in 78% of patients. At the end of all cTACE, objective response and non-progressive disease were 32% and 70%, respectively. Progression-free survival was 18.3 ± 13.3 months (median 14.9) and median overall survival (OS) from start of treatment was 74 months. The OS at 1 year, 2 years, 3 years and 5 years was 91% (IC = 84–99%), 84% (CI = 72–95%), 69% (CI = 53–84%) and 63% (C = 46–81%), respectively. Conclusions: cTACE using streptozocin is an effective and well-tolerated palliative option for patients with neuroendocrine liver metastases, associated with prolonged survival and delayed time to progression.

## 1. Introduction

Neuroendocrine tumors (NETs) are neoplasms derived from neuroendocrine cells with the property to synthesize peptide hormones and sometimes biologically active substances responsible for carcinoid syndrome [1]. About 10% of patients with gastroenteropancreatic NETs develop carcinoid syndrome with flushing and diarrhea [2].

The most common anatomical sites of origin are the gastrointestinal tract, the pancreas and, more rarely, the lungs [3]. These rare tumors have a prevalence estimated to be 35/100,000 and an incidence of 2.5–5.3/100,000 population [4]. The incidence appears to be increasing [5].

One of the most important factors affecting patient survival is the presence of liver metastases [6]. Unfortunately, NETs are frequently metastatic at the time of diagnosis, and the liver is the most vulnerable site of metastases [7].

A high liver tumor burden is another negative prognostic factor, a low tumor burden being associated with prolonged survival and favorable treatment response [8,9].

In nonsurgical candidates, treatment is indicated to control hormone-related symptoms refractory to medical management, including pain or diarrhea, to control tumor growth and to improve survival [10].

Somatostatin analogues are offered as a first-line therapy since somatostatin inhibits both secretion by, and the growth of, many different types of neuroendocrine tumors [11]. Somatostatin analogues can improve symptoms with a good tolerance but tumor response is widely variable [11]. Local treatments such as thermal ablation or loco-regional therapies including trans-catheter arterial embolization (TAE) or conventional chemoembolization (cTACE) may then be performed [12,13]. TAE and cTACE are favored in patients with multiple liver metastases [14].

TAE and cTACE consist in the intravascular delivery of chemotherapeutic and/or embolic agents. Arterial embolization is associated with ischemic necrosis of target tumors with the occlusion of arterial blood supply [15]. cTACE combines the effects of chemotherapy injected directly within the tumor arterial feeders with those of anoxia induced by embolization [16,17,18,19]. Intra-arterial therapies are particularly relevant because neuroendocrine liver metastases are highly vascular, supplied by hepatic artery branches.

Several cTACE techniques have been reported, but there are no data suggesting superiority of one technique over the other. The mechanism of action for cTACE is the selective obstruction of tumor-feeding arteries by injection of chemotherapeutic agents mixed with Lipiodol followed by injection of embolization particles. The use of different drugs including doxorubicin or streptozocin has been reported [13,14,16,17,18,19]. Drug-eluting beads (DEBs) loaded with doxorubicin have also been occasionally used [20,21]. One study has suggested that the use of streptozocin was associated with a higher rate of disease control compared to doxorubicin [22].

## 2. Objectives

The aim of our study was to evaluate the tolerability and clinical, biological and radiological tumor response and survival rates in patients with unresectable neuroendocrine liver metastases treated by cTACE using streptozocin, Lipiodol and embolization microspheres.

## 3. Methods

This was an observational retrospective monocentric study to evaluate the efficacy and tolerability of cTACE in patients presenting with unresectable liver metastases. All consecutive patients who were treated with cTACE using a combination of streptozocin, iodinized oil (Lipiodol) and tris-acryl microspheres between March 2010 and March 2020 were included. The study was approved by the Institutional Review Board (reference 1783). Patients were older than 18 years, had normal or moderately impaired hepatic function (bilirubin level, serum AST and ALT less than 3 times the upper limit of normal), normal or moderately impaired renal function (creatinine clearance > 30 mL/min calculated using the MDRD formula) and normal coagulation parameters (prothrombin time > 50% and platelet count > 50,000). The disease was predominant to the liver although extrahepatic disease was not an exclusion criterion. Patients treated with chemotherapeutic agents in their history before the first cTACE were not excluded.

Exclusion criteria consisted in patients with severely impaired hepatic and renal function, patients with dominant extrahepatic disease (lung, bone and lymph-node metastases), and poor performance status according to the Eastern Cooperative Oncology Group scale (ECOG status 3–4).

Indication for cTACE was discussed during our pluridisciplinary tumor board. cTACE was indicated to control tumor growth and/or hormone-related symptoms such as carcinoid syndrome despite the use of somatostatin analogues.

Baseline characteristics included age and body mass index (BMI) at the time of first cTACE, Ki67 tumor index, grade, differentiation, location of primary tumor (resected or not), synchronous or metachronous liver metastases, previous treatments, hepatic tumor burden, and the presence or absence of extrahepatic metastases. Biological tests included tumoral markers, chromogranin A and urinary 5-hydroxyindoleacetic acid (5-HIAA), liver enzymes (AST, ALT, total bilirubin level) and creatinine clearance.

Radiological evaluation was based on a tri-phasic iodinated contrast-enhanced (unenhanced, arterial and portal phases) CT or MRI. The tumor burden was evaluated by a visual semi-quantitative analysis demonstrated to have good inter- and intra-observer agreements [23]. The thresholds for the tumor classes (<10%, 10–25%, 25–50%, 50–75%, >75%) already used in other studies were chosen [23,24].

cTACE was performed via the femoral artery. Selective catheterization of the celiac artery was performed using different shapes of 5-F catheters and superselective catheterization of the proper hepatic artery using a 2.7-F microcatheter (Progreat, Terumo, Japan) successively.

An initial celiac trunk arteriography was obtained to evaluate the distribution of hepatic arteries. Selective angiography of the superior mesenteric and left gastric arteries was obtained if an anatomical variation had been identified on CT or MRI.

The technique consisted in intra-arterial injection of an emulsion of 1.5 g of streptozocin (Zanosar, Keocyt Riemser, France) in 7.5 mL solution mixed in 10–20 mL of iodized oil (Lipiodol, Guerbet, France). The amount of iodized oil was based on tumor volume and vascularity on initial angiography. Complimentary embolization using 100–300 or 300–500 µm tris-acryl microspheres (Embosphere, Merit Medical, South Jordan, UT, USA) was performed to occlude the 2nd–3rd-order hepatic artery branches. The volume of Lipiodol and the size of embolization microspheres were chosen based on angiographic appearance. More Lipiodol was used in cases of large hypervascular nodules. Smaller microspheres were selected to embolize small branches.

Intravenous hydration was started before cTACE and patients were premedicated with antibiotics and antiemetic drugs. Prophylactic subcutaneous administration of somatostatin analogues was applied for the prevention of carcinoid crisis at the time of treatment. General anesthesia was induced before administration of chemotherapy as the acidity of the drug is associated with intractable pain at the time of intra-arterial administration. Morphine was then given in the immediate post-embolization period.

To reduce the risk and severity of complications, in patients with a high tumor burden, a sequential approach was favored, with 2 to 4 procedures to treat the whole liver. The most affected lobe was treated first. The second session was performed 4–6 weeks later. In these patients, sessions were considered as a single processing cycle and imaging assessment was obtained after treatment of the whole liver.

Adverse events and complications related to the treatment were recorded using Common Terminology Criteria for Adverse Events (CTCAE v 5.0), with a special focus on post-embolization syndrome (fever, abdominal pain, nausea, vomiting) and serious adverse events during admission and the next thirty days. Changes in liver enzymes (AST, ALT, bilirubin) and creatinine clearance at day 1 and day 5 and 2 months after cTACE were also carefully recorded using the Classification of Cancer Therapy Evaluation Program (CTEP).

Radiological tumor response was evaluated by a tri-phasic iodinated contrast-enhanced CT or MRI every 2–3 months after the start of treatment using the same modality as before treatment.

Tumor response was evaluated for the treated metastases according to the modified Response Evaluation Criteria in Solid Tumors (mRECIST) criteria, which consider tumor response as a decrease in the hypervascularized portion of the lesion at the arterial phase. Complete response (CR) was defined as the complete disappearance of all treated liver metastases, partial response (PR) was defined as at least a 30% reduction in the sum of the largest diameters of the viable portion up to two target lesions. Progressive disease (PD) was defined at least a 20% increase in the sum of the largest diameters of the viable portion or the emergence of new lesions. Stable disease (SD) was defined as a no progression or response disease. Objective response (OR) corresponded to CR + PR, disease control rate (DCR) to CR + PR + SD.

Clinical response was evaluated by the progress of carcinoid symptoms, if present, and biological response by changes in marker levels (chromogranin A in particular). 

Overall survival was evaluated from diagnosis to death and from treatment to death. 

Statistical analyses were performed using Statistica (Tibco^®^ Software; 13.4.0.14; Palo Alto, CA, USA). Baseline demographics and quantitative data were expressed as mean ± standard deviation; median and range were provided if relevant. 

Qualitative data were expressed as frequency and percent.

Progression-free survival (PFS) was defined as time elapsed between first cTACE procedure and date of radiological progression identified on imaging, or censoring.

Overall survival (OS) was measured from time of primary diagnosis and time of start of cTACE treatments. OS was calculated by Kaplan–Meier method, and comparisons were made by log-rank test. Differences were considered statistically significant if the *p* value was less than 0.05. The t test was used to compare means of paired samples. 

## 4. Results

Baseline clinical and radiological features are summarized in Table 1.

Mean age of the 52 patients was 63.8 ± 10.4 years (Table 1). Primary tumor location was GI tract in 29 patients (56%) and pancreas in 14 patients (27%). Most patients (98%) had grade 1 or 2 tumors, and synchronous (79%) liver metastases. Metastases was diffuse in 83% of patients; 29% had half of the liver or more invaded by metastases. Associated extra-hepatic liver metastases were present in 38% of patients but liver was the dominant metastatic site. Symptoms suggestive of carcinoid syndrome were encountered in 80% of patients (42/52). Hepatic resection, systemic chemotherapy and somatostatin analogues were used in 23%, 27% and 71% of patients, respectively. Mean time from initial diagnosis of NETs to diagnosis of liver involvement was 71.6 ± 56.8 months (median 50.4 months, range 12.0–167.4). Mean time from initial diagnosis of liver metastases to first cTACE was 29.5 ± 40.5 months (median 11.9 months, range 0.9–226.9).

A total of 127 cTACE procedures was performed during the study period. Mean number per patient was 2.4 ± 1.3 (median 2.0, range 1–7). During the first cTACE, the whole liver was treated in 18 patients (35%). For the first treatment two, three or four sessions were required for a sequential approach in 20 (38%), 12 (23%) and 2 (4%) patients, respectively. Overall a sequential approach was favored in 34 patients (65%). Additional sessions were required in cases of liver progression.

During the first cTACE, 1.5 g of streptozocin was used in all except 5 patients (1.0 g) because of arterial reflux. Mean Lipiodol volume was 14.4 ± 4.2 mL (median 15.0, range 0–20). The mean volume of microspheres was 2.8 ± 1.6 mL (median 2.0, range 0–8). In 6 patients, no embolization was performed because of stasis, or gelatin sponge was used for arterio-venous shunts.

Post-embolization syndrome was present in 40/52 (77%) patients after first cTACE, 28/37 (76%) patients after second cTACE and 20/22 (90%) after a third cTACE, including patients with abdominal pain who required morphine administration. Overall, post-embolization syndrome was present in 78% of cases.

One patient died 3 days after cTACE from a cardiac complication corresponding to cTACE-related mortality of 2%. Another patient died from an unrelated cause (a ruptured brain aneurysm) the day after a fifth cTACE. Thus, 30-day mortality was 4% and there was a total of 10 serious adverse effects out of 127 (8%) procedures including two deaths (1.6%). All side effects after cTACE sessions are reported in Table 2.

Two patients also treated with peptide receptor radionuclide therapy with ^177^Lu-labelled peptides developed necrosis of biliary ducts with dilatation treated by percutaneous biliary drainage and stenting.

Serologic toxicity graded using the CTEP classification is summarized in Table 3.

Liver enzymes increased significantly from baseline to day 1 post cTACE (*p* < 0.01). At 5 days, a significant decrease was noted (*p* < 0.01). Two months after the procedure, there was no significant difference with the baseline (*p* = 0.06 for AST, 0.29 for ALT and 0.75 for bilirubin). 

No significant change in creatinine clearance was found after cTACE at any time point. No factor associated with complication, including volume of Lipiodol and volume of microspheres, was identified.

After the first whole-liver treatment, 29/42, 69% of patients had resolution of their carcinoid syndrome (*p* = 0.01).

A decrease in the level of chromogranin A was found after treatment although not statistically significant (*p* = 0.053). 

After the first whole-liver treatment, 49 patients were assessed, with 2 patients lost to follow-up and one dead at 3 days. CR, PR and OR rates were 34%, 50% and 84%, respectively. Disease control rate was 96%. OR rates in patients with GI and pancreatic disease were 81.4% and 86%, respectively, without significant difference. At the end of all cTACE, 32% of patients had an OR and 38% SD, resulting in a DCR of 70%.

Mean follow-up was 30.4 ± 20.4 months (median 26.0, range 3.1–83.0). PFS was 18.3 ± 13.3 months (median 14.9, range 2.8–58.2).

Eleven patients had extra-hepatic progression after 20.8 ± 13.3 months (median 18.0, range 7.1–50.1). Figure 1 demonstrates the OS from different time points.

Median OS from primary diagnosis was 158 months (CI 95% = 94–248 months).

Median OS from initiation of treatment such as somatostatin analogues was 74 months (CI 95% = 47 – months).

From initiation of cTACE, the OS at 1 year, 2 years, 3 years and 5 years was 91% (CI = 84–99%), 84% (CI = 72–95%), 69% (CI = 53–84%) and 63% (CI = 46–81%), respectively.

In univariate analysis, no statistically significant difference in OS was found based on the primary location (GI vs. pancreatic, *p* = 0.29), surgical resection of primary tumor (*p* = 0.6), grade (*p* = 0.24) or liver tumor burden less than 50% vs. more than 50% (*p* = 0.69) (Figure 2). A significant difference in median OS was found between patients who had an extra-hepatic metastatic disease before the start of cTACE and those who did not (66 versus 74 months, *p* = 0.039) (Figure 3).

Patients who had multiple cTACE procedures survived longer with median OS of 28 months for patients who have had only one cycle of treatment, versus 67 months for patients treated with at least four sessions (*p* = 0.027) (Figure 4).

## 5. Discussion

NETs are frequently associated with liver metastases at the time of diagnosis. cTACE is one of the reference palliative treatment in patients with uncontrolled carcinoid syndrome or progressive disease [16]. There is a relative variability in the choice of chemotherapeutics drugs. Streptozocin may be considered in cTACE instead of doxorubicin [22]. In their multivariate analysis, Marrache, et al. showed that the use of streptozocin was a predictor of tumor response compared to doxorubicin [22]. In another study, the authors demonstrated that streptozocin was an effective drug for cTACE, safely administered multiples times and inducing durable response and disease control [8]. After first cTACE treatment, the authors found that 43% of patients experienced OR with a DCR rate of 81%. We found even higher figures with 84% of OR and 96% of DCR.

At the end of all cTACE treatments, Dhir, et al. reported 23% of patients with OR, and a DCR rate of 70% using RECIST 1.1 [8]. We reported a sustained OR of 32% at the end of treatment with the same DCR rate (70%). In another study, at the end of treatments, the authors reported an OR in 52% of patients and a DCR of 80% using RECIST 1.1 [25]. Our initial tumor response rates are high, potentially because of the sequential strategy (higher cumulative dose of drug) (Figure 5 and Figure 6). However, after the last cTACE, our results are in the same range as those reported in previous studies with streptozocin using RECIST 1.1 and WHO criteria, respectively [22,26]. Our results are better than in other studies using doxorubicin or mitomycin C, with a DCR rate of 65% and 61% using WHO and RECIST 1.1, respectively [27,28]. Moreover, OR rate was more favorable in our study compared to others using mitomycin C or DEBs loaded with doxorubicin [20,28].

Lipiodol has interesting characteristics compared to DEBs (arterio-portal passage, tumor selectivity). Higher symptomatic response can be expected with cTACE compared to DEB-TACE [29]. In addition, liver necrosis has been reported with the use of DEBs, particularly if there was biliary or portal damage initially [30].

The median PFS in our study was 14.9 months, similar to the results reported by Marrache, et al. [22]. Dhir, et al. reported a higher median PFS of 29.7 months [8]. In other studies with doxorubicin using RECIST 1.1, the authors reported lower median PFS of 10 months [14], 13.8 months [21], or 14 and 12 months (respectively, for small bowel and gastropancreatic NETs) [17]. In one study, no significant efficacy or survival differences were found between transarterial embolization and cTACE [14].

Our median OS was 74 months from the start of cTACE, which was better than in most published series in the literature, with OS ranging from 13.5 to 38 months for cTACE using doxorubicin [14,27]. In studies reporting the use of streptozocin, OS ranged from 44 to 61 months [8,22]. As opposed to previous publications, we have not identified significant differences in OS based on tumor burden and the primary location [8]. Apart from the presence of extra-hepatic disease at initiation of cTACE, we were unable to demonstrate any other independent prognostic factor for survival. As already reported, cTACE is effective in controlling carcinoid syndrome allowing the dose of somatostatin analogues to be reduced, which is one of the main benefits for affected patients [22,25,28]. A potential advantage of the sequential strategy is to reduce systemic toxicity of streptozocin and treatment-related complications. No case of renal failure was reported although multiples sessions were performed. Transient systemic serologic toxicity affecting liver enzymes was reported [25,31]. We report similar complications and mortality rates to those of other studies with 4% of mortality at 30 days [23,25].

Recent studies advocate the use of selective internal radiation therapy (SIRT) with Y90 as a potential alternative to cTACE to treat liver metastases [32,33]. Safety and efficacy of SIRT were demonstrated in patients with disease progression after prior therapies and in those with high liver tumor burden [30,32]. Of interest, Do Minh, et al. reported better OS and PFS using cTACE as compared to SIRT [34]. OS and PFS were 33.8 versus 23.6 months and 21.6 versus 11.2 months after cTACE versus SIRT, respectively [34]. The authors reported similar rates of adverse events [34].

Our study had several limitations, including the small number of patients (NETs are rare neoplasms), the retrospective design of the study and the absence of a control group. It should be emphasized that it is difficult to conduct a randomized trial in patients with a rare disease, to compare different drugs or intra-arterial therapies.

## 6. Conclusions

cTACE using streptozocin is an effective and well-tolerated palliative option for patients with unresectable neuroendocrine liver metastases, which can be repeated and induces durable response and disease control. Based on published studies, a prospective comparison of embolization, chemoembolization, radioembolization and drug-eluting bead chemoembolization should be conducted.

## Figures and Tables

**Figure 1 cancers-15-04021-f001:**
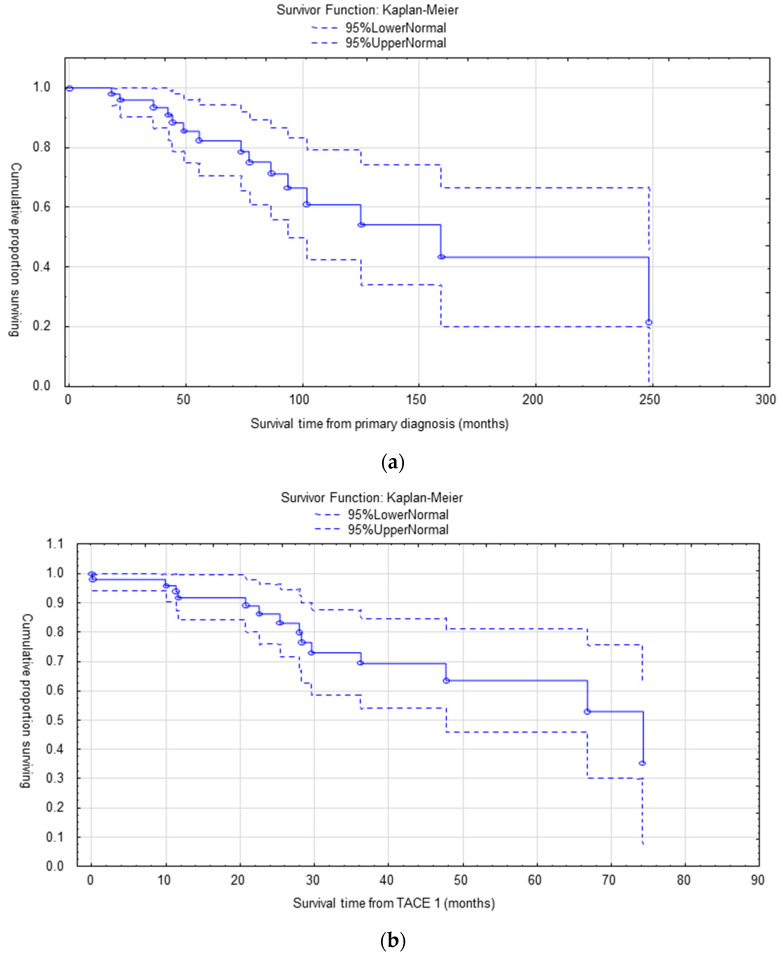
(**a**) Median OS from primary diagnosis was 158 months (95% CI = 94–248). (**b**) Median OS from start of treatment was 74 months (95% CI = 47–).

**Figure 2 cancers-15-04021-f002:**
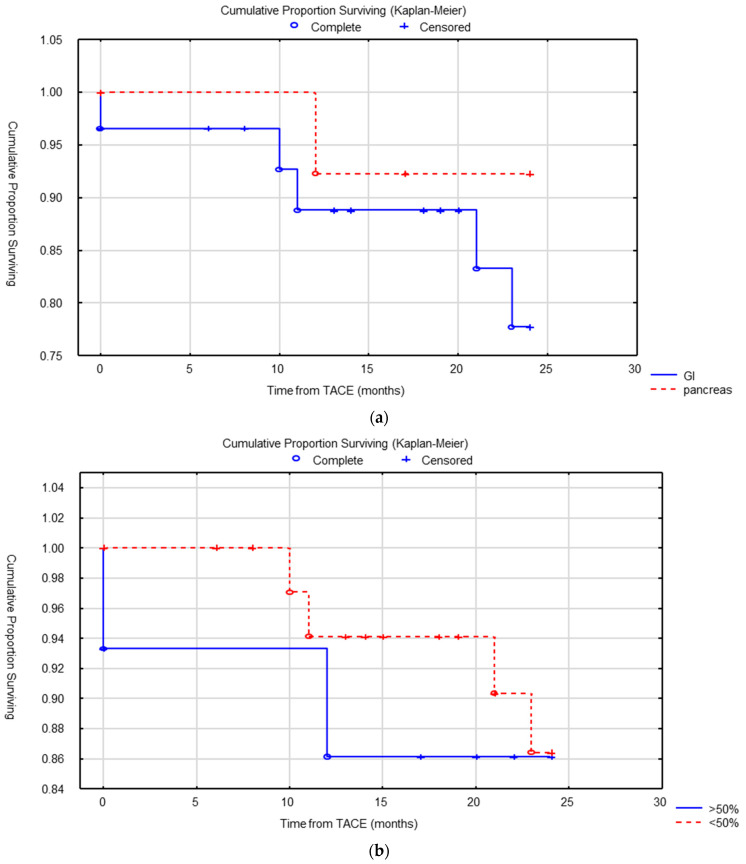
(**a**) OS according to primary tumor location (gastrointestinal GI vs. pancreas) was not different (log-rank test) (*p* = 0.29). (**b**) OS according to liver tumor burden (less than 50% vs. more than 50% liver replacement) was not different (*p* = 0.79).

**Figure 3 cancers-15-04021-f003:**
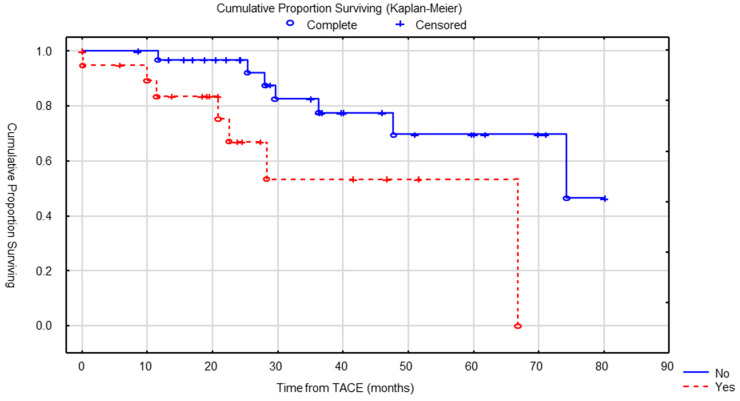
OS according to the presence of extra-hepatic disease (no vs. yes) at the time of first cTACE was significantly different (*p* = 0.039).

**Figure 4 cancers-15-04021-f004:**
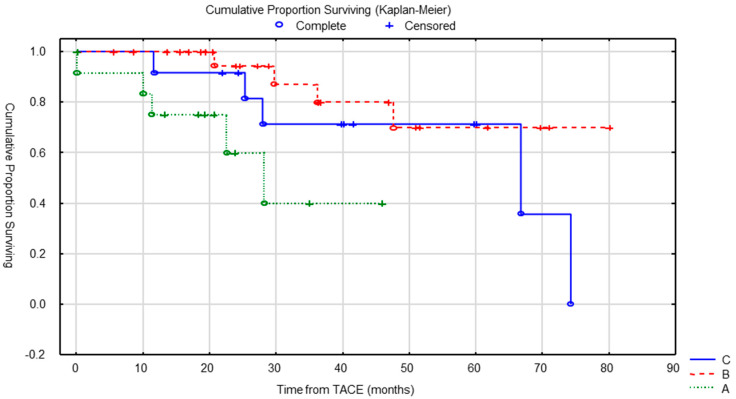
OS according to the number of cTACE was significantly different (*p* = 0.027). A = 1 treatment, B = 2–3 treatments and C = 4 or more treatments.

**Figure 5 cancers-15-04021-f005:**
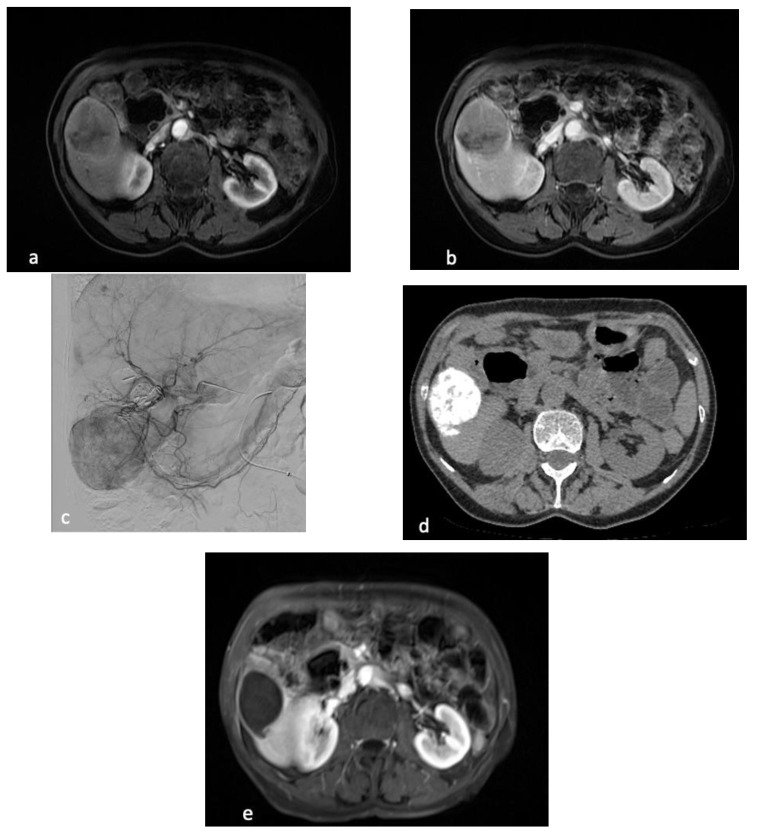
77-year old female presenting with ileal neuroendocrine liver metastases predominant in right liver with 10–25% tumor burden. Axial contrast-enhanced MRI at the arterial phase (**a**) and portal phase (**b**), demonstrates the largest target lesion located in segment VI. Angiographic evaluation (**c**) during the first session of cTACE confirms the tumoral hypervascularization. Additional smaller bilobar nodules were present. Axial reconstruction of CT obtained 2 months after cTACE (**d**) demonstrates complete Lipiodol uptake. Contrast-enhanced MRI post cTACE (**e**) confirms the absence of residual enhancement of the lesion corresponding to complete response.

**Figure 6 cancers-15-04021-f006:**
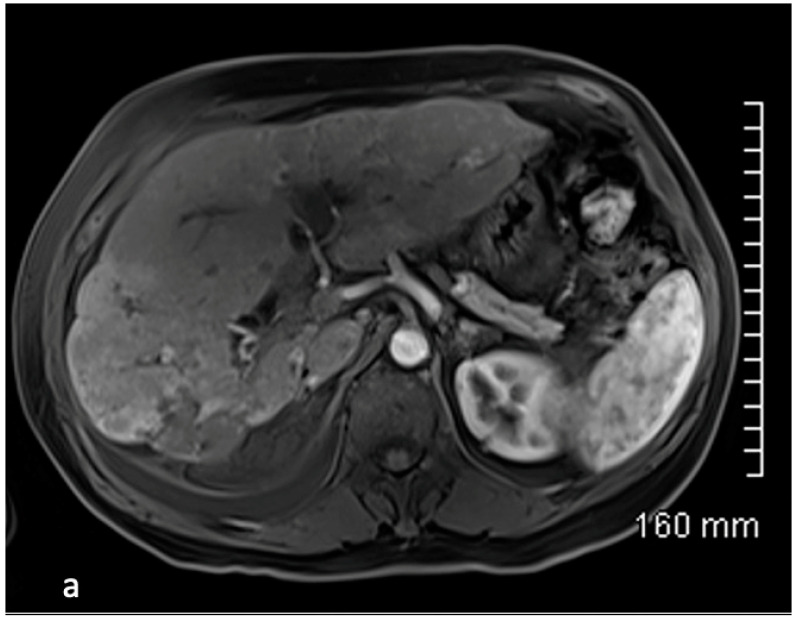
54-year-old female with ileal NET G2 tumor treated by sequential chemoembolization using Streptozocin with a partial response for 14 months. Contrast-enhanced MRI demonstrates the presence of small hypervascular liver metastases mainly identified at the arterial phase mostly right liver (**a**). T2-weighted (**b**) and diffusion-weighted (**c**) MRI sequences demonstrate a significant bi-lobar liver progression with diffuse metastatic infiltration of the liver parenchyma (>50%). A second session of cTACE was indicated. Angiographic evaluation (**d**) confirms the diffuse metastatic infiltration of the liver.

**Table 1 cancers-15-04021-t001:** Baseline characteristics.

	n = 52
Gender, male/female n	25/27
Mean age (years)	
Age at diagnosis (years)	61.0 ± 11.0 (med 63.0, min–max 31.9–80.8)
Age at first TACE (years)	63.8 ± 10.4 (med 65.1, 33.3–82.6)
**Body mass index (BMI) n = 50**	24.6 ± 4.5 (med 23.6, 17.2–36.0)
BMI ≤ 20, n (%)	7 (14)
20 ≤ BMI ≤ 25, n (%)	23 (46)
25 ≤ BMI ≤ 30, n (%)	12 (24)
BMI > 30, n (%)	8 (16)
**Primary tumour, n (%)**	
GI tract	29 (56)
Pancreas	14 (27)
Lung	4 (8)
Unknown	5 (8)
Surgical resection	30 (58)
**Grade, n (%) (n = 49)**	
Well-differentiated (grade 1)	26 (53)
Grade 2	22 (45)
Grade 3	1 (2)
Mean Ki67 index (n = 40)	5.7 ± 7.1 (med 2.0, min–max 0–30)
**Liver metastases, n (%)**	
Synchronous liver metastases	41 (79)
Metachronous liver metastases	11 (21)
**Previous treatment, n (%) ***	
Hepatic surgery	12 (23)
Radiofrequency ablation	2 (4)
Systemic chemotherapy	14 (27)
Somatostatin analogues	37 (71)
TACE	8 (15)
Peptide receptor radionuclide therapy	2 (4)
**Liver tumour burden, n (%)**	
<10	9 (17)
10–25	15 (29)
25–50	13 (25)
50–75	8 (16)
>75%	7 (13)
**Extrahepatic disease**	20 (38)
**Discovery mode, n (%)**	
Carcinoid syndrome	12 (23)
Abdominal or lumbar pain	22 (42)
Fortuitous	11 (21)
Other	4 (8)
Unknown	3 (6)
**Carcinoid dominant symptoms, n (%)**	
Diarrhea	29 (56)
Flushes	23 (44)
**Baseline Chromogranin A (n = 45)**	1030.1 ± 1644.6 (med 350.0, min–max 25.0–7629.0)
ng/mL	
Elevated (>100) n (%)	37 (82)
Normal	8 (18)
**Urinary 5-HIAA (n = 20)**	70.0 ± 92.6 (med 35.5, 3.4–280.0)
µmol/24 h	
Elevated (>50) n (%)	6 (30)
Normal (<50)	14 (70)
**Baseline liver and renal function (n = 51)**	
AST (mean ± SD, normal < 35), IU/L	29.6 ± 20.0 (med 25.0, min–max 12.0–144.0)
ALT (mean ± SD, normal < 45), IU/L	31.1 ± 31.3 (med 20.0, min–max 7.0–216.0)
Bilirubin (mean ± SD, normal < 16) IU/L	11.4 ± 4.7 (med 10.0, min–max 5.0–31.0)
Creatinine clearance (MDRD) mL/min	80.3 ± 18.8 (85, 26–117)
Clearance >60 mL/min,	44 (86)
Clearance 30–60 mL/min,	7 (14)
**Radiological features of liver metastases**	
Longest diameter (mean SD) mm	
Target 1 (n = 52)	55.9 ± 35.1 (48.5, 10–158.0)
Target 2 (n = 43)	34.1 ± 22.8 (30.0, 6.0–100.0)
Sum of longest disease (mean SD)	84.1 ± 50.9 (77.0, 12.0–195.0)

* percentages > 100% since patients may have received multiple treatments.

**Table 2 cancers-15-04021-t002:** Side effects of all TACE.

List of Side Effects	n = 127
Post embolization syndrome	99 (78%)
Serious adverse events	10 (8%)
Pneumopathy	2 (1.6%)
Heart failure	2 (1.6%)
Bowel obstruction	2 (1.6%)
ARDS	1 (0.8%)
Liver abscess	1 (0.8%)
Death	2 (1.6%)

ARDS = acute respiratory distress syndrome.

**Table 3 cancers-15-04021-t003:** Serologic toxicity.

Toxicity	Normal	Grade 1	Grade 2	Grade 3	Grade 4
**AST (IU/L)**		**(>ULN-2.5× ULN)**	**(2.5–5× ULN)**	**(5–20× ULN)**	**(>20× ULN)**
Baseline (n = 52)	40 (77%)	11 (21%)	1 (2%)	0 (0%)	0 (0%)
Day 1 (n = 51)	0 (0%)	3 (6%)	3 (6%)	20 (39%)	25 (49%)
Day 5 (n = 49)	0 (0%)	11 (23%)	10 (20%)	24 (49%)	4 (8%)
At 2 months (n = 38)	25 (66%)	10 (26%)	3 (8%)	0 (0%)	0 (0%)
**ALT (IU/L)**		**(>ULN-2.5× ULN)**	**(2.5–5× ULN)**	**(5–20× ULN)**	**(>20× ULN)**
Baseline (n = 52)	45 (86%)	6 (12%)	1 (2%)	0 (0%)	0 (0%)
Day 1 (n = 51)	2 (4%)	8 (16%)	4 (8%)	19 (37%)	18 (35%)
Day 5 (n = 49)	3 (6%)	9 (19%)	7 (14%)	25 (51%)	5 (10%)
At 2 months (n = 38)	27 (71%)	10 (26%)	1 (3%)	0 (0%)	0 (0%)
**Bilirubin (IU/L)**		**(>ULN-1.5× ULN)**	**(1.5–3× ULN)**	**(3–10× ULN)**	**(>10× ULN)**
Baseline (n = 52)	45 (86%)	6 (12%)	1 (2%)	0 (0%)	0 (0%)
Day 1 (n = 51)	16 (31%)	20 (39%)	9 (18%)	6 (12%)	0 (0%)
Day 5 (n = 49)	16 (32%)	15 (31%)	13 (27%)	5 (10%)	0 (0%)
At 2 months (n = 36)	32 (89%)	4 (11%)	0 (0%)	0 (0%)	0 (0%)

ULN = upper limit of normal. Normal values: AST < 35 IU/L, ALT < 45 IU/L, total bilirubin < 16 IU/L. Numbers in bold indicate the impact of chemoembolization on the liver.

## Data Availability

The data that supports the findings of our study are available from the corresponding author upon reasonable request.

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
