# Peer review of "Conventional Transarterial Chemo embolization Using Streptozocin in Patients with Unresectable Neuroendocrine Liver Metastases"

_cancers, 2023, doi:10.3390/cancers15164021_

Round 1

Reviewer 1 Report

From clinical standpoint of view this is a important manuscript because transarterial chemoembolization of liver metastases in pts. with NET has a proven place in the management of these patients. To improve the understanding of the treatment technique as well as the results for the readers, I suggest to consider the following recommendations:

1. Methods
How was the water in oil-emulsion of Streptozocin and Lipiodol prepared? Did you prepare in every patient an emulsion of 20 cc Lipiodol plus 7,5 ml Streptozocin at the beginning of the procedure or did you consider the number and size of tumors in advance and prepared in patients with lower tumor load 7,5 ml Streptozocin with a smaller volume of Lipiodol?

2. Patients
You mentioned that the indication for TACE was either to control tumor growth and/or to control hormone-related symptoms which were present despite treatment with Somatostatin analogoues. Please indicate the numbers of patients with these specific indications.

3. Results
Please indicate the cumulative dose of Streptozocin in the treated patient population. In the discussion section you mentioned that this was probably an important reason for better results in your study group campare to other repost.
You indicated that the range of Lipiodol used during TACE was 0-20 cc. Did that mean that some patients did not recieve any Lipiodol? If that is the case you have to report the number of these patients and the reasons why they did not receive Lipiodol. This information should also be included in the methods section and in the abstract.

4. Discussion
You compared the results of treatments in your patients with the results of reports in the literature in terms of local response. In your study you used the mRECIST system. However to make these comparisons sufficient you should also indicate the response evaluations systems used in the cited reports. If RECIST classification or WHO classification was used in the other reports the results are nor directly comparable. 

5. Figures number 1 and 2
Please indicate the total survival time of these patinets from the time point of first TACE.

no comments

Author Response

Dear editor,

Please find enclosed our response to reviewer’s comment.

Haut du formulaire

 Reviewer 1

From clinical standpoint of view this is a important manuscript because transarterial chemoembolization of liver metastases in pts. with NET has a proven place in the management of these patients. To improve the understanding of the treatment technique as well as the results for the readers, I suggest to consider the following recommendations:

  1. Methods

How was the water in oil-emulsion of Streptozocin and Lipiodol prepared? Did you prepare in every patient an emulsion of 20 cc Lipiodol plus 7,5 ml Streptozocin at the beginning of the procedure or did you consider the number and size of tumors in advance and prepared in patients with lower tumor load 7,5 ml Streptozocin with a smaller volume of Lipiodol?

As suggested by the reviewer, the technique chapter has been detailed in the revised version of the manuscript as follows:

In the Methods chapter

“An initial celiac trunk arteriography was obtained to evaluate the distribution of hepatic arteries. An initial celiac trunk arteriography was obtained to evaluate the distribution of hepatic arteries. Selective angiography of the superior mesenteric and left gastric arteries was obtained if an anatomical variation had been identified on CT or MRI.

The technique consisted in intra-arterial injection of an emulsion of 1.5g of streptozocin (Zanosar, Keocyt Riemser, France) in 7.5 mL solution mixed in 10-20mL of iodized oil (Lipiodol, Guerbet, France). The amount of iodized oil was based on tumor volume and vascularity on initial angiography. Complimentary embolization using 100-300 or 300-500 µm tris-acryl microspheres (Embosphere, Merit Medical, USA) was performed to occlude the 2nd-3nd order hepatic artery branches. The volume of Lipiodol and the size of embolization microspheres were chosen based on angiographic appearance. More Lipiodol was used in case of large hypervascular nodules. Smaller microspheres were selected to embolize small branches.

Intravenous hydratation was started before cTACE and patients were premedicated with antibiotics and antiemetic drugs. Prophylactic subcutaneous administration of somatostatin analogues was applied for the prevention of carcinoid crisis at the time of treatment. General anesthesia was induced before administration of chemotherapy. The acidity of the drug is associated with intractable pain at the time of intra-arterial administration. Morphine was then given in the immediate post-embolization period.

To reduce the risk and severity of complications, in patients with a high tumor burden, a sequential approach was favored, with 2 to 4 procedures to treat the whole liver.”

2.Patients
You mentioned that the indication for TACE was either to control tumor growth and/or to control hormone-related symptoms which were present despite treatment with Somatostatin analogoues. Please indicate the numbers of patients with these specific indications.

As suggested by the reviewer, the numbers of patients with these specific indications was added in the revised version of the manuscript as follows:

In the Methods chapter

“Indication for cTACE was discussed during our pluridisciplinary tumor board. cTACE was indicated to control tumor growth and/or hormone-related symptoms such as carcinoid syndrome despite the use of somatostatin analogues. Exclusion criteria consisted in patients with severely impaired hepatic and renal function, patients with dominant extrahepatic disease (lung, bone, and lymph node metastases), and poor performance status according to the Eastern Cooperative Oncology Group scale (ECOG status 3-4).”

In the results chapter

“Primary tumor location was GI tract in 29 patients (56%) and pancreas in 14 patients (27%). Most patients (98%) had grade 1 and 2 tumors, and synchronous (79%) liver metastases. Metastases was diffuse in 83% of patients, 29% had half of the liver or more invaded by metastases. Associated extra-hepatic liver metastases were present in 38% of patients but liver was the dominant metastatic site. Symptoms suggestive of carcinoid syndrome were encountered in 80% of patients (42/52).”

  1. Results
    Please indicate the cumulative dose of Streptozocin in the treated patient population. In the discussion section you mentioned that this was probably an important reason for better results in your study group campare to other repost.
    You indicated that the range of Lipiodol used during TACE was 0-20 cc. Did that mean that some patients did not recieve any Lipiodol? If that is the case you have to report the number of these patients and the reasons why they did not receive Lipiodol. This information should also be included in the methods section and in the abstract.

As suggested by the reviewer, details of the technique and doses of Streptozocin have been added to the revised version of the manuscript as follows

In the Methods chapter

« The technique consisted in intra-arterial injection of an emulsion of 1.5g of streptozocin (Zanosar, Keocyt Riemser, France) in 7.5 mL solution mixed in 10-20mL of iodized oil (Lipiodol, Guerbet, France). The amount of iodized oil was based on tumor volume and vascularity on initial angiography. Complimentary embolization using 100-300 or 300-500 µm tris-acryl microspheres (Embosphere, Merit Medical, USA) was performed to occlude the 2nd-3nd order hepatic artery branches. »

In the Results chapter

« All patients had received Streptozocin and Lipiodol. During the first cTACE, 1.5g of streptozocin was used in all except 5 patients (1.0g) because of arterial reflux. Mean lipiodol volume was 14.4 ±4.2 mL (median 15.0, range 0-20). The mean volume of microspheres was 2.8 ±1.6 mL (median 2.0, range 0-8). In 6 patients, no embolization was performed because of stasis or gelatin sponge was used or arterio-venous shunts. »

« No significant change in creatinine clearance was found after cTACE at any time point. No factor associated with complication was identified including volume of lipiodol and volume of microspheres.”

  1. Discussion
    You compared the results of treatments in your patients with the results of reports in the literature in terms of local response. In your study you used the mRECIST system. However to make these comparisons sufficient you should also indicate the response evaluations systems used in the cited reports. If RECIST classification or WHO classification was used in the other reports the results are nor directly comparable. 

As suggested by the reviewer, this chapter has been modified in the revised version of the manuscript as follows

in the Method chapter “Tumor response was evaluated for the treated metastases according to the modified Response Evaluation Criteria in Solid Tumors (mRECIST) criteria, which consider tumor response as a decrease in the hypervascularized portion of the lesion at the arterial phase”

into the discussion:

 “At the end of all cTACE treatments, Dhir, et al reported 23% of patients with OR, and DCR rate of 70% using RECIST 1.1  [8]. We reported a sustained OR of 32% at the end of treatment with the same DCR rate (70%). In another study, at the end of treatments, the authors reported an OR in 52% of patients and DCR of 80% using RECIST 1.1 [25]. Our initial tumor response rates are high potentially because of the sequential strategy (higher cumulative dose of drug) (Fig. 5 and 6). However, after the last cTACE, our results are in the same range as those reported in previous studies with streptozocin using RECIST 1.1 and WHO criteria, respectively [22,26]. Our results are better than in other studies using doxorubicin or mitomycin C, with DCR rate of 65% and 61% using WHO and RECIST 1.1, respectively [27,28]. Moreover, OR rate was more favorable in our study compared to others using Mitomycin C or DEB loaded with doxorubicin [20,28].

Lipiodol has interesting characteristics compared to DEB (arterio-portal passage, tumor selectivity). Higher symptomatic response can be expected with cTACE compared to DEB-TACE [29]. In addition, liver necrosis has been reported with the use of DEB, particularly if there was biliary or portal damage initially [30].

The median PFS in our study was 14.9 months, similar to the results reported by Marrache, et al. [22], Dhir, et al reported a higher median PFS of 29.7 months [8]. In other studies with doxorubicin using RECIST 1.1, the authors reported lower median PFS of 10 months [14], 13.8 months [21], or 14 and 12 months (respectively for small bowel and gastropancreatic NETs) [17].”

  1. Figures number 1 and 2
    Please indicate the total survival time of these patinets from the time point of first TACE.

As suggested by the reviewer, the total survival time of the patients from the time of first TACE was added in the revised version of the manuscript as follows

Figure 1. (b) Median OS from start of treatment was 74 months (95% CI=47-).

Figure 2. median OS from start of treatment according to primary tumor location (a) and to liver tumor burden (b)

Reviewer 2 Report

The purpose of this study was to evaluate the clinical, biological and radiological response and tolerability of conventional Transarterial Chemoembolization (cTACE) using streptozocin for unresectable neuroendocrine liver metastases. Patients and Methods: A total of 52   patients with predominant liver disease were treated with cTACE using an emulsion of streptozocin, Lipiodol and embolization particles. This is a clinical study. 

1. The consent for clinical study from the grouped patients should be provided.

2.  The conclusion section should be revised. In conclusion, it is generally necessary to state: theoretical value (the problems explained by the research results and the principles reflected), practical value (the role played by the research results in practical application and its significance), the advantages and disadvantages compared with the existing research results, the development and improvement made, the unresolved problems in the research, and the relevant suggestions and prospects.

3. Authors should add a blank before and after “+” and “=”. Authors are advised to revise the layout of figure 2.

4. A related reference on TAE should be cited and discussed to enrich the background of this paper (Development of PVA-based microsphere as a potential embolization agent).

Minor editing of English language required.

Author Response

The purpose of this study was to evaluate the clinical, biological and radiological response and tolerability of conventional Transarterial Chemoembolization (cTACE) using streptozocin for unresectable neuroendocrine liver metastases. Patients and Methods: A total of 52   patients with predominant liver disease were treated with cTACE using an emulsion of streptozocin, Lipiodol and embolization particles. This is a clinical study. 

  1. The consent for clinical study from the grouped patients should be provided.

As suggested by the reviewer, the consent for clinical study was added in the revised version of the manuscript as follows :

In the Methods chapter

« The study was approved by the Institutional Review board (reference 1783). »

  1. The conclusion section should be revised. In conclusion, it is generally necessary to state: theoretical value (the problems explained by the research results and the principles reflected), practical value (the role played by the research results in practical application and its significance), the advantages and disadvantages compared with the existing research results, the development and improvement made, the unresolved problems in the research, and the relevant suggestions and prospects.

as suggested by the reviewer, the conclusion has been modified in the revised version of the manuscript as follows

“cTACE using streptozocin is an effective and well tolerated palliative option for patients with unresectable neuroendocrine liver metastases, which can be repeated and induces durable response and disease control. Based on published studies, a prospective comparison of embolization, chemoembolization, radioembolization, and drug-eluting bead chemoembolization should be conducted.”

  1. Authors should add a blank before and after “+” and “=”. Authors are advised to revise the layout of figure 2.

As suggested by the reviewer, a blank before and after « + »  and « = » wad added in the revised version of the manuscript as well as the layout of figure 2

  1. A related reference on TAE should be cited and discussed to enrich the background of this paper (Development of PVA-based microsphere as a potential embolization agent). TAE results added in the text.

as suggested by the reviewer, a related reference on TAE was cited in the reviser version of the manuscript, as follows

in the Introduction

“TAE and cTACE consist in the intravascular delivery of chemotherapeutic and/or embolic agents. Arterial embolization is associated with ischemic necrosis of target tumors with the occlusion of arterial blood supply [15]. cTACE combines the effects of chemotherapy injected directly within the tumor arterial feeders with those of anoxia induced by embolization [16-19]. Intra-arterial therapies are particularly relevant because neuroendocrine liver metastases are highly vascular, supplied by hepatic artery branches.

Even if several cTACE techniques have been reported, but there are no data suggesting superiority of one technique over the other. The mechanism of action for cTACE is the selective obstruction of tumor-feeding arteries by injection of chemotherapeutic agents mixed with Lipiodol followed by injection of embolization particles. The use of different drugs including doxorubicin or streptozocin has been reported [13,14,16-19]. Drug-eluting beads (DEB) loaded with doxorubicin have also been occasionally used [20,21]. One study has suggested that the use of streptozocin was associated with a higher rate of disease control compared to doxorubicin [22].”

in the Discussion

« In one study, no significant efficacy and survival differences were found between transarterial embolization and cTACE [14]. »